# Nodal Expansion, Tumor Infiltration and Exhaustion of Neoepitope-Specific Th Cells After Prophylactic Peptide Vaccination and Anti-CTLA4 Therapy in Mouse Melanoma B16

**DOI:** 10.3390/ijms26136453

**Published:** 2025-07-04

**Authors:** Alexandra V. Shabalkina, Anna V. Izosimova, Ekaterina O. Ryzhichenko, Elizaveta V. Shurganova, Daria S. Myalik, Sofia V. Maryanchik, Valeria K. Ruppel, Dmitriy I. Knyazev, Nadezhda R. Khilal, Ekaterina V. Barsova, Irina A. Shagina, George V. Sharonov

**Affiliations:** 1Research Institute of Translational Medicine, Pirogov Russian National Research Medical University, 1 Ostrovityanova, Moscow 117997, Russia; shabalkina_av@rsmu.ru (A.V.S.); sofyamaryanchik@gmail.com (S.V.M.); ruppel_vk@rsmu.ru (V.K.R.); barsova_ev@rsmu.ru (E.V.B.); sharina_ia@rsmu.ru (I.A.S.); 2Department of Genomics of Adaptive Immunity, Shemyakin-Ovchinnikov Institute of Bioorganic Chemistry RAS, 16/10 Miklukho-Maklaya, Moscow 117997, Russia; ryzhichenko_eo@rsmu.ru; 3Research Institute of Experimental Oncology and Translational Biomedicine, Privolzhsky Research Medical University, 10/1 Minin and Pozharsky Sq., Nizhny Novgorod 603950, Russia; izosimova_a@pimunn.net (A.V.I.); liza2309shur@yandex.ru (E.V.S.); zavyalova_d@pimunn.net (D.S.M.); 4Pathoanatomical Department, Nizhny Novgorod Regional Clinical Cancer Hospital, Nizhny Novgorod 603126, Russia; 5Molecular Genetics Laboratory, The University Clinic, Privolzhsky Research Medical University, 10/1 Minin and Pozharsky Sq., Nizhny Novgorod 603950, Russia; knyazev_d@pimunn.net (D.I.K.); khilal_n@pimunn.net (N.R.K.)

**Keywords:** peptide anticancer vaccine, B16 melanoma, TCR repertoire

## Abstract

Peptide vaccines possess several advantages over mRNA vaccines but are generally less effective at inducing antitumor immunity. The bottlenecks limiting peptide vaccine efficacy could be elucidated by tracking and comparing vaccine-induced T-lymphocytes in successful and unsuccessful cases. Here we have applied our recent database of neoantigen-specific T cell receptors (TCRs) to profile tumor-specific T cells following vaccination with a neoantigen peptide vaccine and to correlate this with the response. Mice were vaccinated prophylactically with p30 peptide encoding B16 melanoma neoantigen (K739N mutation in *Kif18b* gene). The B16F0 melanoma in the vaccinated mice was additionally treated by a CTLA-4 checkpoint blockade. T cells from the tumors, tumor-draining lymph nodes (tdLNs) and vaccine depots were isolated, phenotyped, sorted by subsets and sequenced for TCR repertoires. The vaccine induced the accumulation of tumor-specific CD4+ Th cells in the tdLNs, while in the tumors these cells were present and their frequencies were not changed by the vaccine. These cells also accumulated at the vaccine depots, where they were phenotypically skewed by the vaccine components; however, these effects were minor due to approximately 50-fold lower cell quantities compared to the tdLNs. Only some of the p30-specific Th cells showed tumoricidal activity, as revealed by the reverse correlation of their frequencies in the tdLNs with the tumor size. The CTLA-4 blockade did not affect the tumor growth or the frequencies of tumor-specific cells but did stimulate Th cell motility. Thus, we have shown that tumor-specific Th clones accumulate and/or expand in the tdLNs, which correlates with tumor suppression but only for some of these clones. Tumor infiltration by these clones is not correlated with the growth rate.

## 1. Introduction

Neoantigen-targeted mRNA vaccines have recently demonstrated outstanding efficiency in treating melanoma both in mice and humans [1,2,3,4]. Peptide vaccines generally have lower efficiency. For example, p30 antigen gives complete remission in the mRNA form, while in the peptide form it delays tumor growth just by 20–25% [1,2]. However, peptide vaccines have several advantages over mRNA vaccines. They are easier to produce and handle and they allow the use of modified aminoacids and protection from proteolysis [5]. The use of altered peptide sequences, different adjuvants and vaccination schemes can dramatically change vaccine effects; however, there is still no rationale for the proper combining of these components in such a multivariable setting [6]. The tracking of antigen-specific response and its type, spatiotemporal properties and magnitude would uncover bottlenecks of peptide anticancer vaccines and improve their properties.

Antibody titers are the main readouts for vaccine efficiency used in development and in practice. However, there is a long route between antigen administration and antibody production by plasma cells. Moreover, the role of antibodies for anticancer immunity is controversial [7]. This complicates rational anticancer vaccine development because there are no distinct intermediate checkpoints that can be used to explain vaccine activity. Tumor infiltration by T cells and their phenotypic markers could possibly serve as readouts of vaccine efficiency, but this is complicated by indistinguishability between antigen-specific and off-target effects of vaccines.

Off-target vaccine activity comes mainly from adjuvants that boost innate immunity, which also triggers memory T cells specific to allergens and persistent infections [8]. Exaggerated nonspecific inflammation is particularly undesired for cancer vaccines since it drives immune exhaustion and tumor microenvironment resistance [9]. The crucial point that defines vaccination outcome is the type and the level of tumor-specific response [4,10]. Methods for dissecting tumor-specific responses and T cell receptors (TCRs) are rapidly evolving; however, they remain quite sophisticated [11,12,13,14,15,16,17,18,19]. As a result, the existing data on the phenotype, number and residence of tumor-specific lymphocytes is still very sparse.

In our previous study we showed that the clonality of CD8 lymphocytes and local enrichment of tissue-relevant convergent clonotypes of public TCRs in tumor-draining lymph nodes are hallmarks of effective antitumor vaccination [20]. However, CD8+ T cells act on the late effector phase of antitumor immunity, while CD4 cells orchestrate the initial decision-making stage of adaptive immune response [21]. It was also shown that vaccines and immune checkpoint blockade can elicit antitumor immunity through CD4+ T cells without engaging CD8+ T cells [2,22]. Accordingly, vaccination with 27-mer immunogenic B16 neoepitope peptides from [1,2] elicited an antigen-specific response of CD4+ Th cells but not of CD8+ T cells [19].

In the current study we aimed to profile the distribution of neoantigen-specific CD4+ T cells (Th and Tregs) between the vaccination site, tdLNs and the tumor in relation to the tumor suppression. We have used our recently elaborated database of TCRs specific to several immunogenic neoepitopes of B16 melanoma including p30 peptide [19]. This database provides a convenient tool for deciphering the CD4+ T cell response on a B16 mouse melanoma model. In this study we vaccinated mice with p30 neoantigen peptide to skew antitumor immunity towards certain epitopes and make it trackable by known TCRs. The mice were also treated with immune checkpoint blockade (ICB), which is commonly applied after tumor vaccination in humans [3,4,23]. We have shown that p30 vaccination does increase antigen-specific clones in tdLNs by several orders of magnitude. The frequencies of most but not all of these clones were not correlated with the tumor size. However, some clones appeared tumoricidal, which was revealed by their inverse correlation with the tumor size. In tumors these clones were also present in the controls (adjuvant-only) mice at comparable frequencies. This indicates that the tumoricidal activity of neoepitope-specific Th cells is released mainly in tdLNs, presumably by the epitope spreading mechanism. Tumoricidal clones were also most public among others, highlighting the potential importance of eliciting public clones by antitumor vaccines.

## 2. Results

### 2.1. Peptide Neoantigen Vaccines Suppress Tumor Growth in Fraction of Mice

Transgenic mice bearing the FoxP3-GFP chimeric protein [24] were prophylactically vaccinated with the p30 neoantigenic peptide vaccine in order to skew the further antitumor immune response towards certain epitopes. For the first vaccination, p30 peptide (50 µg per mice) was formulated in complete Freund adjuvant (CFA) and administered subcutaneously (s.c.) into both flanks and on the back near the tail (50 µL of CFA/PBS emulsion each). The boosting vaccination was performed by peptide (50 µg per mice) and was administered in PolyI:C adjuvant (Figure 1). After the first vaccination there were persistent CFA plaques at the sites of the injections (Figure 2A). A nonpersisting vaccine formulation for the boosting dose is commonly used in combination with CFA in order to (i) spread specific cells over the body and prevent their local dysfunction and (ii) avoid the possible morbidity of repetitive injury at the vaccination sites [25].

The prophylactic vaccination with p30 peptide decreased the tumor growth, measured as mean tumor size (Figure 1B). If the tumor growth curves were analyzed for individual mice, an apparent subdivision of the mice into two groups was present for the control mice (adjuvant-only group) and became more obvious for the mice vaccinated with the p30 peptide (Figure 1C). After vaccination, the number of mice in each group changed but not significantly (3/7 before and 4/5 mice after therapy with small/large tumors, respectively); however, the mean tumor volume curves for each group differed significantly (Figure 1D). This indicates that the observed separation of the mice by tumor size in the vaccinated group reflects the individual mouse tolerance to the tumor rather than a variable response to the vaccine.

The anti-CTLA-4 (aCTLA4) therapy with blocking only antibodies (clone 9D9) of the vaccinated mice did not affect the tumor growth compared to the p30 vaccine-only group (Figure 1B). Yet, the aCTLA4 therapy increased the dispersion in tumor volume, which degraded the difference between the large and small tumors (Figure 1C).

### 2.2. p30 Peptide and aCTLA4 Therapy Decrease Cell Sequestration and Immunosuppression of T- and B-Lymphocytes in CFA Plaques

Persistent vaccine depots are reported to be sites of local accumulation of antigen-specific cells in temporary quazi-LN structures termed tertiary lymphoid structures (TLSs) [26,27]. We found large condensed subcutaneous plaques at the sites of the peptide/CFA injections (Figure 2A). Macroscopically these plaques resembled lymph nodes but are several times larger (Figure 2A). We evaluated whether these CFA plaques could accumulate p30-specific cells and act as TLSs to aid antitumor immunity by analyzing the lymphocyte count, composition and phenotype of the CFA plaques for the different mouse groups (Figure 2).

We found that both the p30 peptide and aCTLA4 therapy significantly altered the size, cell composition and phenotype of the lymphocytes in the CFA plaques. The aCTLA4 therapy plaques were significantly smaller compared to the adjuvant-only p30 + PBS groups, which is reflected as a smaller mean number of CD3+ cells in these groups: 10,894 ± 2380, 12,474 ± 2243 and 2892 ± 790 T cells for the adjuvant-only, p30 + PBS and p30 + aCTLA4 groups, respectively. This effect is related mainly to the Th cells and to a lesser extent to the CD8+ T cells (Figure 2C). B cell accumulation in the CFA plaques was markedly reduced by the peptide antigen and just slightly by the CTLA-4 blockade.

We also analyzed CD38, a common marker of lymphocyte immunosuppression, and CD134, a marker of antigen-specific activation [28,29]. There was a minor increase in CD134+ expressing Th cells with aCTLA4 therapy and decrease in CD134+ CD8+ T cells with p30 peptide (Appendix A). The number of CD83-positive cells was affected substantially by the presence of peptide antigen. CD83+ cells were elevated in Th subset and decreased in CD8+ T cells and B cells (Figure 2D). We did not detect any significant differences in Treg cells between the groups (Appendix A).

### 2.3. aCTLA4 Therapy Stimulates Accumulation of Th Cells in Tumors

The CFA plaque shrinkage and depletion of Th lymphocytes in the course of aCTLA4 therapy possibly reflects that the therapy induces the migration of Th cells to the tumor and/or the tdLNs. We analyzed the T and B cell density and their relative numbers in the tumors. The cell densities were measured as cells per mg of excised tumor mass and were equal between the groups for the CD8+ T- and B-lymphocytes but slightly elevated for Th cells after the peptide vaccine and aCTLA therapy (Figure 3B). As the cell densities are highly diverse between the tumors, we also calculated the relative quantities for these subsets, which are more uniform and indicative. The elevated relative number of Th cells with the aCTLA4 therapy was confirmed by an increased T/B cell ratio, Th/B cell ratio and CD4+ T cell percentage from the CD3+ T cells (Figure 2C). Notably, the frequency of CD134+ from Th cells in the p30 + aCTLA4 group but not in the other treatment groups was correlated with tumor size (Figure 3E, Spearman r = 0.63, *p* < 0.005).

### 2.4. p30-Specific Th Cells Prevail in tdLNs and Have Trends Towards Negative Correlations with Tumor Size

Recently we elaborated a database for the TCR beta-chain (TCRb) specific to B16 neoepitopes (see Section 4), where specific clonotypes are derived as TCRb clusters. Each cluster represents a group of several (from 4 to 200) clonotypes with equal TRBV and CDR3 segments related to the neighboring segments by one aminoacid mismatch. The clusters were selected by their enrichment in the draining lymph nodes after vaccination with a certain peptide compared to 13 other peptides. For the p30 antigen there are nine such clusters.

We analyzed the presence of the clonotypes from p30-specific clusters for Th cells from 13 tdLN samples, 10 CFA plaque samples and 20 tumor samples. The Treg repertoires were analyzed for 10 tdLN samples. The samples were selected according to the sorted cell count since repertoires are not informative for limited cell counts. All nine p30-related clusters were found in the current data.

Eight out of nine of the p30-related clusters were found in the tdLN and plaque samples, with three of them (C14, C25, C26) present in the majority of the samples. In the tdLNs and CFA plaques of the p30 vaccinated mice, the frequencies of the specific TCRb clonotypes was several orders higher compared to the mice vaccinated with adjuvant only (Figure 4A) and constitute up to 10% of the Th cells (C25 cluster in tdLNs). We found no difference in the frequency of either the individual clusters or the sum of all the specific clones between the vaccination groups and between the tdLNs and CFA plaques.

The frequencies of the p30-specific clones in the tdLN/CFA plaque samples showed trends towards a negative correlation with the tumor volume (Figure 4). The most distinctive correlation was for the C14 cluster, with a Spearman r = −0.65 (*p* < 0.005). Other clusters were also trending towards such a correlation but for tumors smaller than 200 mm^3^. This may indicate the activation of immunosuppression mechanisms in large tumors, which prevents specific clones from exerting their effector functions. Treg cells from the p30-specific clusters were also found in the tdLN samples, with average 2.5-times lower frequencies compared to the Th cells. No correlation of the p30-related T-regs with treatment and tumor size was found (Appendix A).

The frequencies of the specific cells for most clusters were indistinctive between the CFA plaques and tdLNs, except for the C26 cluster, which was more frequent in the tdLN samples (Figure 4B). Since the number of lymphocytes in the tdLNs was approx. 50 times higher, this points that the tdLNs are the major sites for the proliferation of antigen-specific cells, while the CFA plaques are likely to be infiltrated by tdLN-generated cells.

### 2.5. p30-Specific Th Clones Present in Tumors Without p30 Vaccination Are Not Correlated with Tumor Growth

The presence of p30-specific clonotypes within tumor-infiltrating Th lymphocytes was analyzed. Clonotypes from specific clusters were found within the tumors with the prevalence of the same three clusters as in the tdLNs (Figure 5). The average frequencies were comparable to that in the tdLNs; however, no correlation with the treatment and tumor size was found. It should be noted that the repertoires were measured only for tumors larger than 100 mm^3^. This, together with the higher dispersion of clone frequencies, likely underlined our failure to derive correlations. For example, clusters C14 and C25 have trends towards a negative correlation with the tumor size, but the observed stochasticity requires more data points to reveal its significance.

Contrary to the tdLNs, p30-specific clones were present in the tumors even without p30 vaccination at comparable frequencies (Figure 5). Notably, the most frequent p30-specific clones with and without p30 vaccination were from the same clusters, C14, C25 and C26 (Figure 4). This is in agreement with the notion that efficient anticancer vaccines often elicit public and frequently assembled TCR clones (see Section 3).

## 3. Discussion

Anticancer vaccination with peptides is known to have moderate efficiency, which is highly dependent on the adjuvant and vaccination scheme [30,31]. IFA is a widely used formulation to provide long-term antigen retention at vaccination sites and elicit both a humoral and cellular response. Earlier we used footpad vaccination with B16 peptide antigens in IFA for the detection of p30-specific clones [19]. However, after a single vaccination, the number of specific clones in the draining popliteal LNs does not exceed 2%. To enhance the presence of specific clones and to skew most of the antitumor specificity towards the p30 antigen, we used vaccination with p30 peptide in CFA following bossing in PolyI:C adjuvants. This allowed us to increase the percentage of p30-specific Th cells in the tdLNs up to an 18% maximum and 3% mean frequencies.

The CFA adjuvant is known to form granulomas, which were shown to retain antigen-specific CD8+ T cells after vaccination with H-2Db–restricted gp100 antigen and to induce their exhaustion [27]. Here we observed the selective retention of Th cells in CFA plaques after vaccination with Th-related p30 peptide antigen (Figure 2C).

However, the number of CFA-sequestered Th cells was approx. 50 times lower than the number of these cells in the tdLNs (Appendix A) and the sequestered cells were not enriched with p30-specific clones (Figure 4B). This is in agreement with previous data showing that antigen-specific CD4+ cells proliferate in LNs and accumulate at the sites of antigen deposition in a non-proliferative state [32]. Moreover, we found that the aCTLA4 therapy substantially decreases this sequestration of Th cells in CFA plaques. Since in this study the CFA plaque-retained fraction was small, this does not have much therapeutic effect, yet this newly discovered activity of the CTLA4 blockade can be used in other therapeutic settings.

As we have shown with p30 peptide, the presence of this antigen could drastically shape the microenvironment at the CFA depot, namely, the cell composition and expression of CD83 on the different lymphocyte subsets (Figure 2). It is reasonable to suppose that these effects are dependent on the antigen, which, in turn, is detected by memory T cells, which preserve their functional phenotype in the altered microenvironment [33]. The lineage of the memory T cell type would therefore be the result of previously encountering the same or a mimicking antigen. It is therefore preferable if tumor-specific T cells share their specificity with virus-specific T cells that usually elicit a Th1 response.

Accordingly, it was shown that public T cell clonotypes are often responsible for efficient tumor eradication [33,34,35,36]. Likewise in our previous study most of the tdLN-enriched public clones were found in the literature and had tumor- or tissue-related specificities [20]. For the p30-specific clonotypes, we found that the C14 cluster includes CDR3 sequences (CASSFG(R/S)QNTLYF) that is highly homologous with public anti-ribonucleoprotein TCR (CASSFGGQNTLYF, the sequence ID in NCBI GeneBank: AFV60381.1), rheumatoid arthritis-associated TCR (CASSFRGQNTLYF, ID: AAV44222.1) and LCMV gp33-specific CD8+ TCR (CASSFGNSQNTLYF, ID: AFR46249.1). CD8+ TCRs were shown to be activated by Th cells with CDR3 (CGARGTGNTGQLYF) from the C22 cluster, which is also found within the anti-ribonucleoprotein TCR repertoire (CGARETGNTGQLYF, ID: AFV60415.1).

At least for one cluster (C14), we found a reverse correlation of its frequencies in the tdLNs with the tumor size, but this was not the case for the tumor-infiltrating lymphocytes. This highlights that the tdLNs are the major source of antitumor immunity. The absence of Th cell enrichment in the tumors indicates that these cells are not major effectors for tumor eradication. The possible effectors in this case are CD8+ cells, which were shown to be activated by the p30 vaccination, presumably by epitope spreading mechanisms [19]. The epitope spreading mechanism is known to take place in the tdLNs and is licensed by antigen-specific Th cells [37]. In the work of Kreiter et al., vaccination with p30 mRNA also significantly enhances the frequencies of CD8+ cells among the tumor-infiltrating CD45+ leukocytes [2]. Strikingly, the depletion of CD8+ lymphocytes in this work did not reduce the vaccination efficacy [2]. This indicates that there are also other effectors. These could be neutrophils and tumor-associated myeloid cells that were shown to be recruited and/or reprogrammed by Th cells to exert a tumor-killing effect [38,39].

The increased relative number of Th cells in the p30 + aCTAL4 group (Figure 3C) did not result in tumor suppression (Figure 1). This preferred infiltration of Th cells is nonspecific since we did not detect increased numbers of tumor-specific Th cells in this group (Figure 5). This may reflect enhanced Th cell motility with the aCTLA4 therapy, which is also reflected in their depletion in the CFA plaques after therapy (Figure 2C). The enhanced motility of the CD134+ Th cells could also explain the relationship between the frequency of these cells and the tumor size (Figure 3F). CD134 marks activated cells, while the activation of Th1 also induced the expression of chemokine receptors [29,40]. Assuming that the CTLA-4 blockade promotes either Th1 linage skewing and/or other mechanisms for increasing motility, these motile Th cells migrate out from tumors. Thus, small tumors are depleted of CD134+ Th cells more readily than large tumors due to the smaller migration distances required.

## 4. Materials and Methods

### 4.1. Mouse Vaccination

The experiments were carried out on transgenic C57Bl/6-FoxP3^EGFP^ 4–6-month-old female mice (kindly provided by Alexander Rudensky, Sloan Kettering Institute, New York, NY, USA). The transgenic mice were generated against the C57Bl/6 genetic background, by knocking in the chimeric construct of eGFP subcloned into the first exon of the FoxP3 gene [24].

p30 peptide (PSKPSFQEFVDWENVSPELNSTDQPFL) was received from Genscript (Rijswijk, The Netherlands), dissolved in DMSO (Sigma-Aldrich, Saint Louis, MO, USA) at 40 mg/mL and stored at −20 °C. Vaccination was carried out 21 and 7 days before the tumor inoculation. The first vaccination was performed with 50 µg of peptide dissolved at 500 µg/mL in PBS and thoroughly mixed with CFA (InvivoGen Inc., San Diego, CA, USA) at a 1:1 ratio. Then, 50 µL of peptide/CFA emulsion (25 µg of peptide per injection) was injected subcutaneously (s.c.) in each flank and on both sides on the back at the tail base. The boosting vaccination was carried out 7 days before the tumor inoculation with PolyI:C adjuvant (InvivoGen). The p30 peptide in PBS at 250 µg/mL was supplemented with 250 µg/mL PolyI:C. Then, 50 µL of the peptide–PolyI:C mixture was injected s.c. into both flanks (50 µg of peptide per mouse).

The control mice were vaccinated using the same scheme with adjuvant only, without the addition of peptide. The number of mice in the groups was 10 in the vaccinated with the adjuvant-only group, 9 in the vaccinated with p30 peptide and treated with PBS group and 21 in the vaccinated with p30 peptide and treated with aCTLA4 antibodies group.

### 4.2. Tumor Engraftment, aCTLA4 Therapy and Tissue Sampling

The tumors were generated by the subcutaneous (s.c.) injection of 5×10^4^ B16F0 cancer cells in 300 μL PBS into the left flank. The B16F0 melanoma cells were grown in DMEM medium (PanEco, Moscow, Russia) supplemented with 10% fetal bovine serum (FBS, Gibco, Frederick, MD, USA), 0.06% L-glutamine, 50 units/mL penicillin and 50 μg/mL streptomycin. The cells were incubated at 37 °C and 5% CO_2_ and passaged 2–3 times per week. Just before the injection, the cells were detached by trypsin, counted and resuspended at a final concentration of 10^6^ cells in 6 mL PBS.

On days 8, 10 and 12 of the tumor growth, the mice were treated with 250 μg anti-CTLA4 (Clone 9D9, #BP0164, Bio X Cell, Lebanon, NH, USA) injected intraperitoneally (i.p.). The tumor sizes were measured with a caliper three times a week. The tumor volume was calculated using the formula V = a × b × 1/2b, where a and b are the larger and smaller lateral dimensions of the tumor.

On days 14, 15 and 16, the mice were euthanized with Isoflurane, and the tumors, inguinal lymph nodes (tdLN) and CFA plaque from the tumor side were scissored out. The tumors were weighed. All the tissue samples were dissected thoroughly with scissors and passed through a 70 µm cell strainer (BD Biosciences, San Jose, CA, USA). Tumors were centrifuged for 30 min at 800× *g* in 40% Percoll solution (GE Healthcare, Chicago, IL, USA) to remove the excess of cell debris and melanin. The cells were washed in PBS, resuspended in 50 μL of RPMI-1640 medium (PanEco) with 0.5% of bovine serum albumin (BSA, PanEco) and processed for antibody staining.

### 4.3. Flow Cytometric Analysis and Sorting

The samples were stained in 50 μL RPMI medium with 0.5% BSA by the following antibody mixture (0.5 μL per sample): CD45-PerCP/Cy5.5, CD3-APC, CD4-BV650 (BD Biosciences), CD3-APC, CD8-APC/H7, CD83-BV421 (all from BD Biosciences), CD134-PE/Cy7 (BioLegend Inc., San Diego, CA, USA) and 0.25 μL biotin-labeled B220/CD45R/B220 (BioLegend). The cells were stained for 90 min in ice and for the last 45 min were supplemented with 0.5 μL of Steptavidin-PE (Jackson Immunoreseach, Ely, United Kingdom). After staining, the cells were diluted by the addition of 200 μL RPMI and analyzed/sorted without washing. Flow cytometric analysis and sorting were performed with FACSAriaIII cell sorter (BD Biosciences) equipped with 405 nm, 488 nm, 561 nm and 633 nm lasers using a 70 μm nozzle. For the RNA isolation and repertoire analysis, the cells were sorted directly in 100 ul of RNA Safer LS reagent (Magen Biotechnology Co., Ltd., Guangzhou, China) and stored at −80 °C before RNA extraction.

The flow cytometric fcs data files were analyzed by FlowJo v.10 software (BD Biosciences). The subset frequencies were analyzed and graphed with GraphPad Prism v. 9 software (GraphPad Software Inc., v.9.4.1, La Jolla, CA, USA). The data are presented as individual data points with median values. For a comparison of the mouse groups, the Brown–Forsythe and Welch ANOVA test with Dunnett T3 correction for multiple testing were used. For the analysis of the relation between two values, a nonparametric Spearman correlation was used, with the evaluation of two-sided *p* values.

### 4.4. TCR Library Preparation and Sequencing

The total RNA was extracted from the cells in RNA Safer LS reagent solutions. The samples were thawed, kept at room temperature for 30 min and pelleted at 4000× *g* for 10 min. The pellets were resuspended in RNAse-free water and centrifuged again (4000× *g* for 10 min). The pellets were resuspended in 300 μL of RTL lysis buffer (Magen Biotechnology) and processed with the HiPure Total RNA Kit (Magen Biotechnology) for RNA extraction. Next, TCRβ cDNA was generated from 20–40 ng of the eluted RNA with a primer for the TRBV constant region and QScribe III revertase (provided by Ekaterina Barsova) for 30 min at 42 °C. the resulting cDNA samples (20 μL) were purified with MagPure A4 magnetic beads (Magen Biotechnology) at a 1:1.5 sample/beads ratio (*v*:*v*) and resuspended in 20 μL 10 mM TE buffer (10 mM Tris/HCL, 1 mM EDTA, pH8.5). Half of the cDNA was used for multiplex PCR of 30 cycles with TRBV-specific forward primers and TRBC-specific reverse primers. All the primers were synthesized by Evrogen (Moscow, Russia). The amplicons were purified using MagPure A4 beads at a 1:1.5 ratio, washed twice with ethanol, resuspended in 50 μL TE buffer, quantified and used at 100 ng for library preparation. The libraries were prepared with the MGIeasy FS Library Prep Set (MGI Tech Co., Ltd., Shenzhen, China) according to the manufacturer’s protocol, starting at the “End repair and A-tailing” stage. The libraries were converted to nanoballs with DNBSEQ-G50RS High-throughput Sequencing Kit and sequenced with DNBSEQ-G50 sequencer (MGI) using paired-end 150 + 150 nt reads and about 25 reads per cell.

### 4.5. TCR Repertoire Analysis

The TCR repertoires were extracted from the FASTQ reads and downsampled using MiXCR v.4.5.0. software (MiLaboratories Inc, San Francisco, CA, USA). In order to avoid sample size-related biases, all the samples were downsampled to 1000 clonotypes. The TCRb repertoires were analyzed for the presence and frequencies of p30-specific clusters [19]. The list TCRb clonotypes from p30-specific clusters can be found in the VDJdb database (https://vdjdb.cdr3.net/, accessed on 16 April 2025) or in the vdjdb-db github repository chunks (https://github.com/antigenomics/vdjdb-db/blob/master/chunks/Mice_TCRs_specific_to_B16_melanoma_neoantigens_(Shagina_et_al_2024).txt, accessed on 16 April 2025). Specific CDR3 sequences were searched for within the current repertoires with the allowance of one aminoacid mismatch (substitution or insertion or deletion) without fixation of the TRBV segment. For searching we used the Levenshtein python C extension. The fequencies of all the matched clonotypes were summed for each cluster and presented in relation to the tumor volume at day 14. The correlation was significant if the Spearman correlation r differed from zero with *p* < 0.05.

## 5. Conclusions

In the current paper we present new details on the anticancer immunity elicited by peptide anticancer vaccine. We used the p30 peptide with the most immunogenic neoepitope among 50 screened candidates and primarily elicited a response of Th cells but not of Treg and CD8+ T cells [1,2,19]. Vaccination with p30 peptide 21 days before (in the current work), 4 days before or on the day of tumor inoculation (in [1]) produced the same tumor suppression rates, so the timing of the first vaccination is not crucial for the vaccine efficiency.

The accumulation of tumor-specific Th cells in tdLNs but not in tumors has practical and fundamental significance. It indicates that tdLNs are preferred over tumors in the search for tumor-specific Th cells/TCRs in the vaccination context. For the dissection of these cells, tdLNs could be compared with non-draining LNs, as we also supposed in a previous work [20]. From the fundamental view, it is important that peptide vaccine efficiency targeted at CD4+ cells does not rely on tumor inflation of activated Th cells. This also implies epitope spread and other cells as effectors, presumably CD8+ T cells.

We found that neoepitope-specific TCRs are equal in their tumor suppression efficiency in spite of having the same target antigen. The determinants for this efficiency are unknown but this highlights the importance of TCRs that are elicited by vaccine. Earlier we have shown that the eliciting of public or persistent pathogen-specific CD8+ T cell clones differentiates efficient peptide vaccine from inefficient vaccine. CD4+ T cells have more diverse repertoires, with lower frequencies of public clones; however, tumor-suppressive TCRs also have more homologous public clonotypes.

Vaccination with p30 peptide and CTLA-4-blocking-only therapy appeared to be unrelated: the efficiency was not enhanced and neither were the frequencies of the tumor-specific clones changed. This differentiates the CTLA-4 blockade from anti-PD1 therapy [41,42] and should be considered for proper combining of these therapeutic approaches.

## Figures and Tables

**Figure 1 ijms-26-06453-f001:**
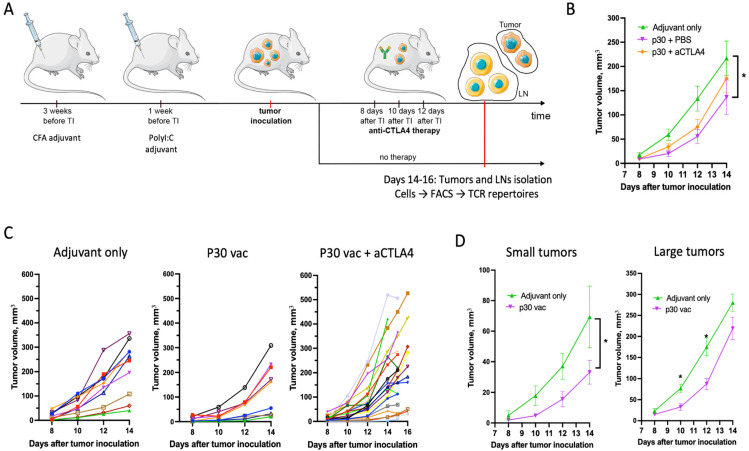
(**A**) Mouse vaccination and treatment scheme. (**B**) Mean tumor growth curves for mice vaccinated with adjuvant only, mice vaccinated with p30 peptide and treated either with PBS or with aCTLA4 antibodies. (**C**) Tumor growth curves for individual mice in each group from (**B**). (**D**) Control mice (adjuvant only) and mice vaccinated with p30 without therapy (p30 + PBS) were subdivided into two groups according to tumor size, which were analyzed separately. In (**B**,**D**), the curves represent mean ± SEM values for each group (*, *p* < 0.05).

**Figure 2 ijms-26-06453-f002:**
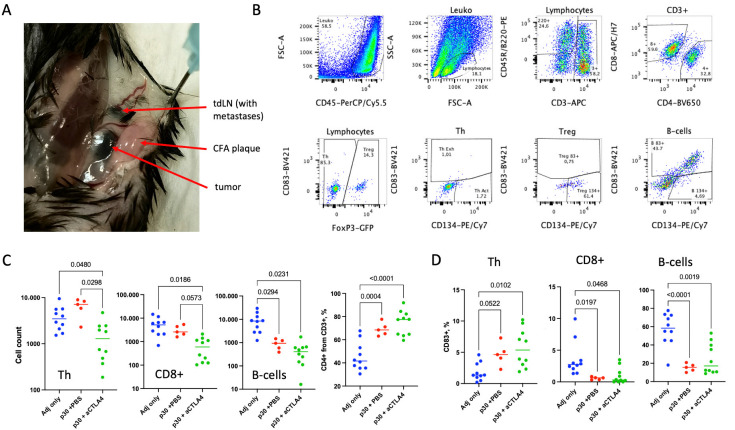
Lymphocyte subset composition and CD83 expression in CFA plaques. (**A**) Typical localization and appearance of the tumor, CFA plaque and tdLN. (**B**) Gating scheme for the analysis of lymphocytes from CFA plaques. Th, Treg, CD8+ T cells and CD45R/B220+ B cells were identified on the basis of expression of CD45, forward and side light scattering (FSC and SSC), CD3, CD45R/B220, CD4, CD8 and FoxP3-GFP. Each subset was analyzed for the percentage of CD83+ and CD134+/CD83− cells. (**C**) Cell counts for Th, CD8+ and B cells in tumor plaques for the three experimental groups: vaccinated with adjacent only (dark blue, Adj only), vaccinated with p30 peptide (red, p30 + PBS) and vaccinated with p30 peptide and treated with aCTLA4 antibodies (green, p30 + CTLA4). (**D**) The percentage of CD83+ cells within Th, CD8+ and B cell subsets. Statistical significance was calculated by Brown–Forsythe and Welch ANOVA test. *p* values below 0.2 are presented on the graphs. *p* values above 0.2 are omitted.

**Figure 3 ijms-26-06453-f003:**
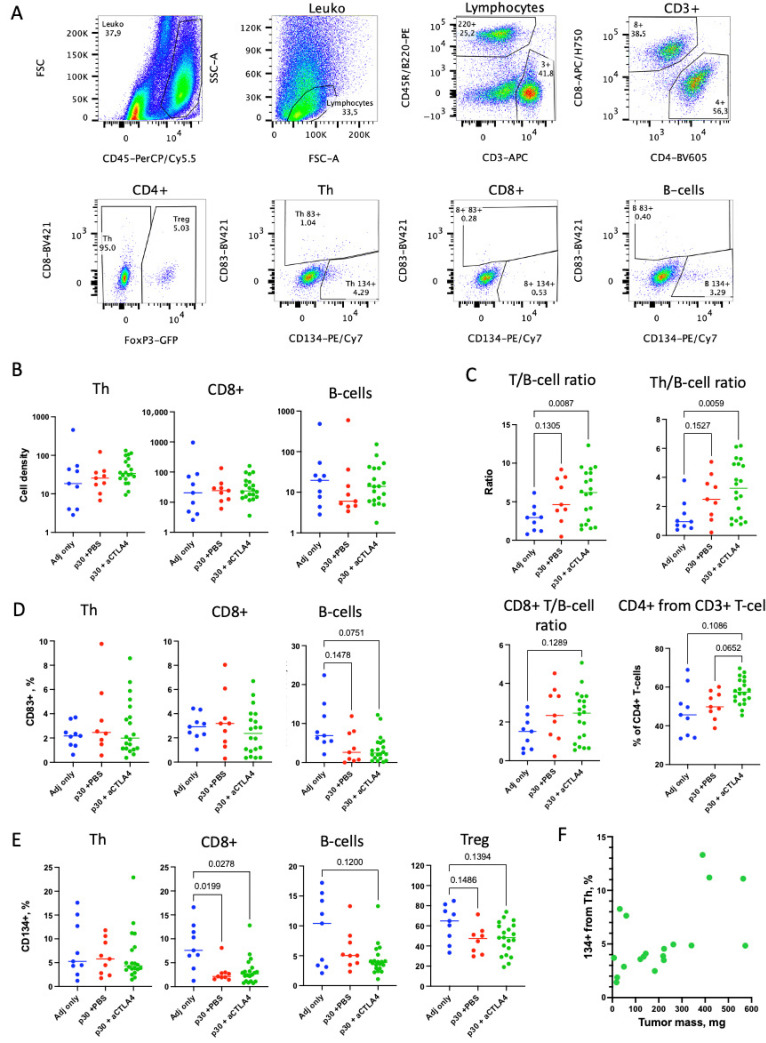
Lymphocyte subset composition and CD83/CD134 expression in tumors. (**A**) Gating scheme for the analysis of lymphocytes from tumors replicated those for CFA plaques (Figure 2) with slightly different population appearance. (**B**) Cell densities (count per mg of tumor) for Th, CD8+ and B cells in tumors for the three experimental groups. Groups are colored as in Figure 2. (**C**) The ratio of T cells to B cells (upper left panel), Th cells to B cells (upper right panel), CD8+ T cells to B cells (lower left panel) and the percentage of CD4+ T cells from all CD3+ T cells (lower right panel) in each mouse. (**D**) The percentage of CD83+ cells within Th, CD8+ and B cell subsets. (**E**) The percentage of CD134+ cells within Th, CD8+, B-and Treg cell subsets. (**F**) Correlation of the percentage of CD134+ from Th cells with tumor mass. Statistical significance was calculated by Brown–Forsythe and Welch ANOVA test. *p* values below 0.2 are presented on the graphs. *p* values above 0.2 are omitted.

**Figure 4 ijms-26-06453-f004:**
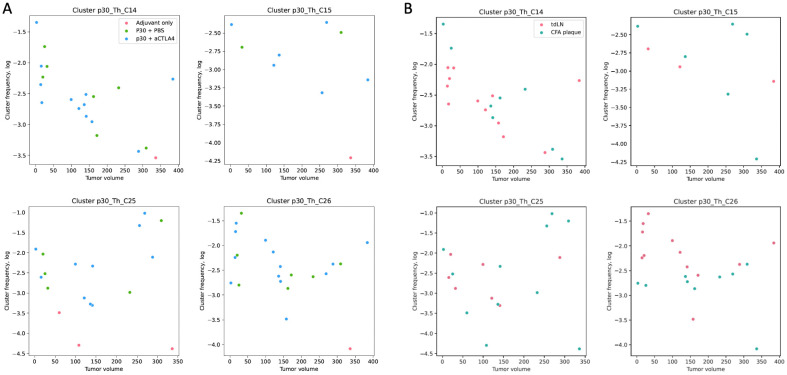
Frequencies of TCRs from most presented p30-specific clusters in CFA plaques and tdLNs depending on the tumor volume. (**A**) Data points are colored according to the treatment group: vaccinated with adjuvant only (red, Adj only), vaccinated with p30 peptide (green, p30 + PBS) and vaccinated with p30 peptide and treated with aCTLA4 antibodies (blue, p30 + CTLA4). (**B**) Data points are colored according to the cell source: tdLN (red) and CFA plaque (green).

**Figure 5 ijms-26-06453-f005:**
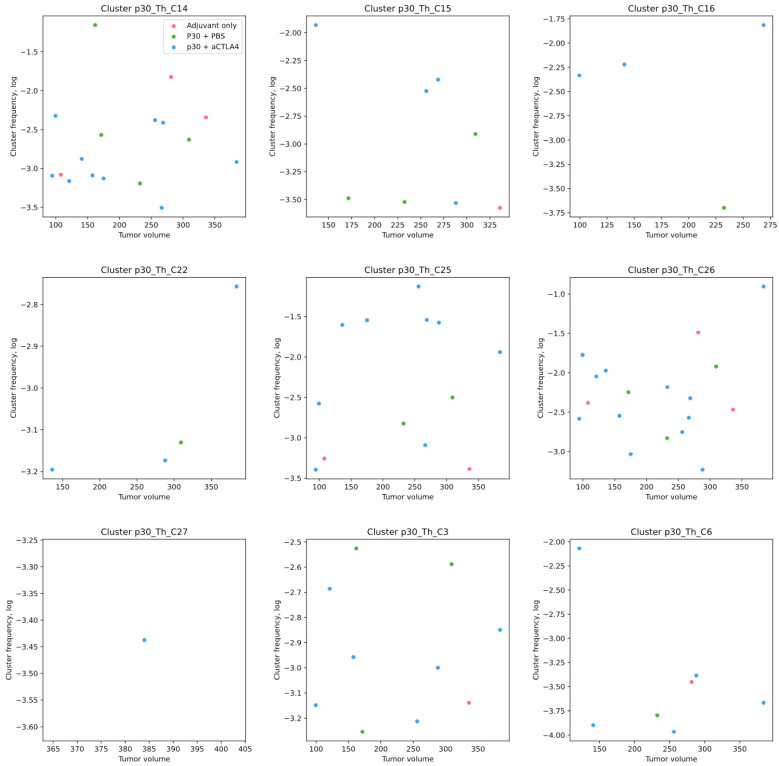
Frequencies of TCRs from p30-specific clusters in tumors depending on the tumor. Data points are colored according to the treatment group: vaccinated with adjuvant only (red, Adj only), vaccinated with p30 peptide (green, p30 + PBS) and vaccinated with p30 peptide and treated with aCTLA4 antibodies (blue, p30 + CTLA4).

## Data Availability

The original contributions presented in this study are included in the article/Appendix A. Further inquiries can be directed to the corresponding author(s).

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
