# Peer review of "Nodal Expansion, Tumor Infiltration and Exhaustion of Neoepitope-Specific Th Cells After Prophylactic Peptide Vaccination and Anti-CTLA4 Therapy in Mouse Melanoma B16"

_ijms, 2025, doi:10.3390/ijms26136453_

Round 1
Reviewer 1 Report
Comments and Suggestions for Authors
Specific comments on weaknesses of the article and what could be improved:
Major points
- The hypothesis in the introduction is not presented clearly. What is the rationale for your study - to compare or to assess effectiveness? In the introduction, there is no clear statement of what is unknown.
- The aim should be clarified, and secondary objectives should be stated (as the results).
- The results are written along with a discussion of the findings. Based on the aim, the results should be presented in a more structured way.
- "We have shown that prophylactic vaccination gives 3% of specific clones among Th cells in tdLNs, which delays tumor outgrowth by only 20%, as in the original study." - Please, explain in the text the biological or clinical relevance of these effects.
- The authors do not clearly state how their findings advance current understanding or how they could influence vaccine design.
Minor points
- In the abstract, "Peptide vaccines possess several advantages over mRNA vaccines but are generally less effective in inducing antitumor immunity." This statement is not true; besides, the authors do not provide sufficient studies and evidence to support it. This statement is contradicted at the beginning of the introduction.
- Please state the study's limitations (e.g., small sample sizes, relevance of the animal model to human disease, or potential confounders (e.g., adjuvant effects, heterogeneity of T-cell response).
- Could you please discuss the clinical implications of the results
- How is this not a conflict of interest? Acknowledgments: The authors would like to acknowledge MDPI redaction office and namely Mr. 467
Albert Li for support in submitting, editing, reviewing, and managing this paper. - What are your recommendations based on your findings?
Author Response
We would like to thank the reviewer for the useful and fair concerns. Suggested corrections do improve the clarity of consistency of the study design and presentation. We appreciate time and effort.
Specific comments on weaknesses of the article and what could be improved:
Major points
Major point 1: The hypothesis in the introduction is not presented clearly. What is the rationale for your study - to compare or to assess effectiveness? In the introduction, there is no clear statement of what is unknown.
Response: The rationale for this study is to reveal the major bottlenecks for efficiency of peptide anticancer vaccines that are generally unknown. Tracking and comparing vaccine-induced T-lymphocytes in responders and non-responders or in active and is the most straightforward way, yet technically challenging. Here we have captured several T-cell profiles in tumor and related lymphoid organs at a single time point. Namely we have measured T-cell quantities, subset composition, exhaustion and activation markers, and frequencies of tumor-specific cells.
To clarify this we have added following statement in the introduction:
- “there is still no rationale for proper combining of these components in such a multivariable setting [6]. Tracking of antigen-specific response, its type, spatiotemporal properties, and magnitude would uncover bottlenecks of peptide anticancer vaccines and improve their properties.”
- “the existing data on the phenotype, number and residence of tumor-specific lymphocytes is still very sparse.”
Major point 2: The aim should be clarified, and secondary objectives should be stated (as the results).
Response: We have clarified the major aim of this in the introduction:
- “In the current study we aimed to profile distribution of neoantigen-specific CD4+ T-cells (Th and Tregs) between vaccination site, tdLNs and tumor with the relation to the tumor suppression.
We have also restructured the Abstract for emphasising the main objectives and for summarising all findings in a more structured way.
Major point 3: The results are written along with a discussion of the findings. Based on the aim, the results should be presented in a more structured way.
Response: Two paragraphs with the discussions were removed from the Results sections 2.2 and 2.3 and were partially rephrased in the Discussion section.
Major point 4: "We have shown that prophylactic vaccination gives 3% of specific clones among Th cells in tdLNs, which delays tumor outgrowth by only 20%, as in the original study." - Please, explain in the text the biological or clinical relevance of these effects.
Response: This part was rewritten to improve clarity:
“Vaccination with p30 peptide 21 days before (in current work), 4 days before or at the day of tumor inoculation (in [1]) gave the same tumor suppression rates”
Major point 5: The authors do not clearly state how their findings advance current understanding or how they could influence vaccine design.
Response: The significance of the current findings for vaccine design was summarized in the rewritten Conclusion section.
Minor points
Minor point 1: In the abstract, "Peptide vaccines possess several advantages over mRNA vaccines but are generally less effective in inducing antitumor immunity." This statement is not true; besides, the authors do not provide sufficient studies and evidence to support it. This statement is contradicted at the beginning of the introduction.
Response: We would not agree with this point. Here are some arguments:
- p30 mRNA vaccine gave complete remission of B16 melanoma in mice while its peptide analog just delayed tumor growth by 20% (references 1,2 from the current paper)
- In metastatic melanoma clinical trials peptide and mRNA vaccines gave comparable 2-year remission rates: 4 of 6 patients in peptide trial (https://doi.org/10.1038/nature22991) and 8 from 13 in mRNA (https://doi.org/10.1038/nature23003). But mRNA study included more patients that makes this study more valuable.
- The most effective COVID vaccine is mRNA vaccine (Pfizer). There are no peptide analogs with nearby efficiency
Minor point 2: Please state the study's limitations (e.g., small sample sizes, relevance of the animal model to human disease, or potential confounders (e.g., adjuvant effects, heterogeneity of T-cell response).
Response: Yes, the study has several limitations and drawbacks.
- We were frustrated by the size of CFA plaques and initially considered them as the major hosts for tumor-specific cells. Due to this about half of tdLNs were ignored and lost.
- Comparison of repertoires between tumor draining and non draining LNs would be beneficial and could shed light on the whole picture of tumor-specific clones. However we did not exercise ndLN due to time limitations. All samples were FACS sorted, which did not allow us to parallel processing.
- 9 clusters of tumor-specific TCRs, their low frequency and high stochasticity requires larger animal groups for more comprehensive analysis and more definitive statements.
Minor point 3: Could you please discuss the clinical implications of the results
Response: We have included possible clinical implications of the current results in the Conclusion section. However we are trying to use careful statements because there is a big gap between mouse melanoma model and patients.
Minor point 4: How is this not a conflict of interest? Acknowledgments: The authors would like to acknowledge MDPI redaction office and namely Mr.
Albert Li for support in submitting, editing, reviewing, and managing this paper.
Response: Possibly yes, it could be a conflict. It was removed.
Minor point 5: What are your recommendations based on your findings?
Response: We summarized our recommendation in the Conclusion section.

Reviewer 2 Report
Comments and Suggestions for Authors
The manuscript by A. V. Shabalkina et al. presents a study evaluating the immune response to prophylactic vaccination with a neoantigen peptide vaccine. Overall, the work is of interest and has potential for publication following minor revisions. I recommend that the authors consider the following comments to strengthen the manuscript:
- The abstract should be rewritten to improve its clarity and appeal. Emphasizing the key findings and the novelty of the study could help in this regard.
- The authors employed polyinosinic:polycytidylic acid [poly(I:C)] to boost vaccination. However, poly(I:C) is known to be associated with toxicity concerns. The authors are encouraged to comment on the rationale behind its use.
- The study was conducted using B16 melanoma model. The inclusion of an additional cancer model could enhance the study's relevance. The authors should justify the model selection.
Author Response
We are grateful to the reviewer for thorough reading of our manuscript and valuable concerns. We tried to do our best to improve it according to the reviewer's suggestions.
The manuscript by A. V. Shabalkina et al. presents a study evaluating the immune response to prophylactic vaccination with a neoantigen peptide vaccine. Overall, the work is of interest and has potential for publication following minor revisions. I recommend that the authors consider the following comments to strengthen the manuscript:
Comment 1:The abstract should be rewritten to improve its clarity and appeal. Emphasizing the key findings and the novelty of the study could help in this regard.
Response: The abstract was rewritten. The major scientific problem and the goal of the current study have been clarified. Key findings were emphasized and clarified.
Comment 2: The authors employed polyinosinic:polycytidylic acid [poly(I:C)] to boost vaccination. However, poly(I:C) is known to be associated with toxicity concerns. The authors are encouraged to comment on the rationale behind its use.
Response: There are several reasons for the use of poly(I:C) to boost vaccination
- At the time of boosting there were clearly palpable paques at sites of first vaccination and there were four such sites. These plaques are rather inflamed and sensitive. So we needed to inject a boosting dose at the higher parts of the flanks.
- Since there were still plaques of CFA we decided that the antigen also still persists at those sites so additional sites of antigen persistence are not required.
- Boosting after Freund adjuvant is commonly performed in non-persistent form (https://doi.org/10.1189/jlb.70.6.849) since effects of boosting occur much faster than of priming.
- Poly(I:C) was efficiently used in mice studies that we encouraged by (ref [2] from the manuscript: https://doi.org/10.1038/nature14426) and also in clinical trails (https://doi.org/10.1038/nature22991). Every adjuvant has adverse events, but for Poly(I:C) it seems to be balanced in the case of cancer.
Comment 3: The study was conducted using B16 melanoma model. The inclusion of an additional cancer model could enhance the study's relevance. The authors should justify the model selection.
Response: The main objective was to profile tumor-specific T-cells in the course of vaccination and therapy. But the only database of tumor-specific clones was done by us for B16 melanoma. Thus within the objecive we are restricted by this tumor model until new databases will be created.

Round 2
Reviewer 1 Report
Comments and Suggestions for Authors
The revision was made according to the suggestions and recommendations. The authors should revise the language and text to reduce plagiarism (which is currently 20%).